# Spatial–Temporal Dynamic Evolution and Influencing Factors of Green Efficiency of Agricultural Water Use in the Yellow River Basin, China

**Weinan Lu** [1,2], **Xinyi Guo** [3], **Wenxin Liu** [2], **Ruirui Du** [2], **Shuyao Chi** [2] and **Boyang Zhou** [1,*]

[1] Centre for Polish Studies, Xi'an International Studies University, Xi'an 710128, China
[2] College of Economics & Management, Northwest A & F University, Yangling 712100, China
[3] Shaanxi Academy of Forestry, Xi'an 710000, China
[*] Correspondence: zhouboyang@xisu.edu.cn

**Abstract:** The progression of agricultural production, ever-increasing industrialization, population boom, and more water-concentrated lifestyles has placed a severe burden on Yellow River Basin's existing water resources, particularly in the current century. In the context of resource and environmental constraints, improving the green efficiency of agricultural water use (AWGE) is an important measure for alleviating the shortage of water resources as well as meeting the intrinsic requirement to promote the green transformation and upgrading of agriculture. This study used the Super Slack-Based Measure (Super-SBM) to measure the AWGE of 87 regions in the Yellow River Basin from 2000 to 2019. Based on spatial and temporal perspectives, it applied Exploratory Spatial Data Analysis (ESDA) to explore the dynamic evolution and regional differences in AWGE. Then, this study used a spatial econometric model to analyze the main factors that influence AWGE in the Yellow River Basin. The results show that, firstly, the AWGE of the Yellow River Basin shows a steady upward trend from 2000 to 2019, but the differences among regions were obvious. Secondly, the AWGE showed an obvious spatial autocorrelation in the Yellow River Basin and showed significant high–high and low–low agglomeration characteristics. Thirdly, rural per capita disposable income and effective irrigation have a positive influence on AWGE, while rural labor transfer, the input intensity of agricultural machinery and water structure have a negative influence. The spatial econometric model regression results show that the influence factors of AWGE in the Yellow River Basin showed significant spatial spillover effects and spatial heterogeneity in their effect. Finally, when improving AWGE in the Yellow River Basin, plans should be formulated according to local conditions. The results of this study can provide new ideas on the study of AWGE in the Yellow River Basin and provide references for the formulation of regional agricultural water resource utilization policies as well.

**Keywords:** agricultural water use; Yellow River Basin; super-SBM model; spatial econometric model

## 1. Introduction

Since the reform and opening up of the country, China's agricultural economy has achieved rapid growth, and a transformation in grain supply has also been realized from a long-term shortage to a basic balance of the total [1,2]. However, the extensive production mode, which relies on long-term, large-scale factor input and the unreasonable disposal of waste, has resulted in there being a great pressure on resource supply and environmental protection [3]. Water resources as a basic element of agricultural production are the most affected [4]. China's agricultural water consumption accounts for 62.32% of total water consumption, but the effective utilization coefficient and water productivity of farmland irrigation water are only 60% of the world's advanced level [5]. Meanwhile, China is in a transition period from traditional agriculture to modern agriculture, and the production mode of agricultural chemicals to promote agricultural economic growth has formed a

serious path dependence [6], which has caused China's agricultural source wastewater pollutant emissions (COD, TN, TP) to remain high for a long time, seriously endangering agricultural ecological security. The structural scarcity of water resources, the deterioration of the water environment, and the aggravation of water ecological risks are intertwined [7], making agricultural water resources one of the main factors restricting the sustainable development of China's agricultural economy.

In the face of the increasingly serious water resource problems, the National Development and Reform Commission of China has also clearly defined the goal of water conservation and efficiency improvement in agricultural development in the National Water Conservation Action Plan issued in 2019 to effectively raise the utilization efficiency and the ability to guarantee water security [8]. The Yellow River, the second longest river in China and the mother river of China, supports 9% of China's population and 15% of China's arable land with about 2% of its runoff. As an important grain base and ecological functional area in China, the status of agricultural water resources in the Yellow River Basin is related to China's grain and ecological security [9]. However, with the rapid economic development and population growth in the Yellow River Basin, the scale of water resource development and utilization along the Yellow River has been expanding, leading to increasingly serious problems such as the outstanding contradiction between the supply and demand of its agricultural water resources and fragile ecological environment [10,11]. In particular, it poses a great threat to agricultural activities within the Yellow River Basin, which is mainly irrigated. Therefore, it represents an important theoretical reference and has practical significance for agricultural water saving and efficiency enhancement, food security protection, and ecological environment improvement in the Yellow River Basin, enabling it to be taken as a research area to explore the spatial–temporal evolution characteristics of agricultural water use efficiency (AWUE) and identify the key factors influencing the improvement in AWUE.

The comprehensive improvement of AWUE relies upon the scientific evaluation and exploration of its development law. Relevant research mainly focuses on three aspects: First, the measurement and evaluation of AWUE, which is essentially a multi-objective and multi-criterion comprehensive issue [12]. Relevant research has been continuously developed and promoted in the process of production and life, and thus, different types of AWUE have been derived. According to its developmental history, the existing research has gradually shifted from research on the efficiency of irrigation water transportation and field utilization engineering to the research on various efficiencies with water resource productivity as the index [13]. Many scholars have also adopted such methods as stochastic frontier function (SFA) [14,15] and data envelopment analysis (DEA) [16–18]. Second, the method is to compare and analyze the distribution dynamics and regional differences in AWUE. Yang et al. [19] and Ma et al. [20] analyzed the regional differences and time-series evolution rules of China's AWUE through a dynamic distribution map and found that it showed a fluctuating upward trend and a distribution trend of "high in the East and low in the west". Third, in terms of the factors influencing AWUE, scholars have discussed its influencing factors from the perspectives of the economic development level, agricultural economic structure, resource endowment, climate environment, technological progress, etc. [21–27].

Referring to existing studies, scholars have carried out rich and meaningful research, but there is still room for improvement and exploration: Firstly, existing studies only measure the AWUE in a region using the efficiency value of a single factor or by only considering the desirable output while neglecting the agricultural non-point source pollution and other undesirable outputs due to negative externalities during the process of agricultural economic growth in the assessment. Secondly, most existing studies take large-scale regions, national, or provincial units, as the research scale. Although the dynamic evolution of the overall AWUE in the region can be controlled from a macro perspective, it is difficult to thoroughly test the differential characteristics of the development of AWUE within the region. The scale dependence of the spatial structure and the important role of

the prefecture-level government in coordinating the allocation of resources within a city make research on intermediate-perspective geographical units scarce, with an urgent need to be improved. Thirdly, the existing research on the factors influencing AWUE is based on the assumption that spatial individuals are homogeneous and unrelated. However, with the increasing opening of the agricultural market economy and the strengthening of regional relevance, the spatial relationship between agricultural production is becoming closer. Neglecting the spatial effect will lead to a deviation in the results.

In view of the above research shortcomings, this paper has made the following three contributions: First, based on the existing research on AWUE, this paper takes agricultural economic output and grain output as desirable outputs, and agricultural carbon emissions and agricultural non-point source pollution as undesirable outputs, and established an evaluation index system of the green efficiency of agricultural water use (AWGE). Second, ESDA was used to analyze and visualize the spatial–temporal dynamic evolution and differentiation characteristics of AWGE in the Yellow River Basin at the city scale. Third, considering the spatial effect of agricultural production, the key factors influencing AWGE in the Yellow River Basin are identified and explored through a spatial econometric model.

## 2. Materials and Methods

### 2.1. Methods

2.1.1. AWGE Evaluation Indicator System

During agricultural production, it is generally desired that the environmental pollution caused by the excessive use of chemicals such as fertilizers, pesticides, and agricultural films be as little as possible, and this output is considered undesirable. The SBM (Slack-Based Measure) model based on unexpected output is a model for measuring efficiency that was first proposed by Tone [28] and it can effectively solve the "crowded" or "slack" of input factors caused by the traditional data envelopment model (DEA) model based on radial and angle. However, the SBM model has the same problem as the traditional DEA model; that is, it is difficult to further distinguish the differences between efficient decision-making units for decision-making units with an efficiency of 1. Based on the SBM model, Tone [29] further defined the Super-SBM model, which is a combination of the Super-DEA model and SBM model that combines the advantages of both models. Compared with the SBM model, the Super-SBM model can further compare and distinguish the efficient decision-making units at the forefront. Drawing on Tone's approach, we first constructed a production possibility set. Assuming that each region is a DMU, there are a total of n DMUs in the production possibility set, with each DUM using $m$ input to produce $r_1$ desirable output and $r_2$ undesirable output. The corresponding vectors are $x \in R_m$, $y^d \in R_{r_1}$, $y^u \in R_{r_2}$, thus defining the matrix:

$$X = [x_1, \ldots, x_k] \in R^{m \times n}, \ Y^d = \left[ Y_1{}^d, \ldots, Y_k^d \right] \in R^{r_1 \times n}, \ Y^u = [y_1^u, \ldots y_k^u] \in R^{r_2 \times n} \quad (1)$$

The model is constructed as follows:

$$\rho = Min \frac{1 - \frac{1}{m} \sum\limits_{i=1}^{m} \left( s_i^- / x_{ik} \right)}{1 + \frac{1}{r_1 + r_2} \left( \sum\limits_{s=1}^{r_1} w_s^d / y_{sk}^d + \sum\limits_{q=1}^{r_2} w_q^u / y_{qk}^u \right)} \quad (2)$$

$$\text{Subject to} \begin{cases} x_{ik} \geq \sum\limits_{j=1,\neq k}^{n} x_{ij}\lambda_j - s_i^-, i=1,2,\ldots\ldots,m, j=1,2,\ldots\ldots,n(j \neq k) \\ y_{sk}^d \leq \sum\limits_{j=1,\neq k}^{n} y_{sj}^d\lambda_j + w_s^d, s=1,2,\ldots\ldots,r_1 \\ y_{qk}^u \geq \sum\limits_{j=1,\neq k}^{n} y_{qj}^u\lambda_j - w_q^u, q=1,2,\ldots\ldots,r_2 \\ \sum\limits_{j=1,j\neq k}^{n} \lambda_j = 1 \\ \lambda_j, s_i^-, w_s^d, w_q^u \geq 0 \end{cases} \tag{3}$$

where, $x_{ik}$, $y_{sk}^d$, $y_{qk}^u$, respectively, represent the input index, desirable output index and undesirable output index. $s_i^-$, $w_s^d$, $w_q^u$ are the slack variables of input, desirable output, and undesirable output, respectively. $\rho$ is the green efficiency of agriculture. $n$ is the number of DMU, $j$ is the $j$th DMU, $k$ represents the $k$th DMU of the current efficiency calculation, and $\lambda_j$ is the $j$th linear combination coefficient of DMU.

In this paper, based on the concept of WUE proposed by Hu et al. [30] and with reference to related studies [5,10], AWGE was defined as the target agricultural water use input to the actual agricultural water use input in the framework of agricultural multifactor production.

The AWGE index is constructed below:

$$AWGE_j^t = \frac{PAW_j^t}{AAW_j^t} = \frac{AAW_j^t - S_{j,w}^t}{AAW_j^t} = 1 - \frac{S_{j,w}^t}{AAW_j^t} \tag{4}$$

where $AWGE_j^t$ represents the AWGE of region $j$ in period $t$. $PAW_j^t$ and $AAW_j^t$ represent the target agricultural water input and the actual agricultural water input of the region $j$ in the period $t$, respectively. $S_{j,w}^t$ represents the slack of agricultural water input of the region $j$ in the period $t$ under the frontier, which is calculated by the Super-SBM model.

2.1.2. Exploratory Spatial Data Analysis Method (ESDA)

ESDA is a collection of spatial data analysis methods and technologies. Its core goal is to test spatial convergence or heterogeneity through global and local spatial auto-correlation measures.

(1)    Global spatial autocorrelation

In this paper, the global Moran I index was adopted to explore the spatial correlation and spatial difference of AWGE among 87 regions in the Yellow River Basin. The calculation formula was as follows:

$$I = \frac{n \sum\limits_{i=1}^{n} \sum\limits_{j=1}^{n} w_{ij}(x_i - \overline{x})(x_j - \overline{x})}{\sum\limits_{i=1}^{n} (x_i - \overline{x}) \sum\limits_{i=1}^{n} \sum\limits_{j=1}^{n} w_{ij}} \tag{5}$$

where $n$ is the sample size; $x_i$, and $x_j$ are the observation quantities of space positions $i$ and $j$; and $w_{ij}$ represents the proximity relationship between spatial positions $i$ and $j$. When $i$ and $j$ are adjacent, $w_{ij} = 1$; additionally, when it is the other way around, it is 0.

(2)    Local Spatial Autocorrelation (LISA)

In this paper, LISA was used to further measure the local spatial variation in AWGE in the Yellow River Basin, and the different degree and significance levels of local spatial agglomeration were analyzed. The calculation formula was as follows:

$$I_i = z_i \sum\nolimits_j w_{ij} z_j, \ z_i = \frac{n(x_i - \overline{x})^2}{\sum_i (x_i - \overline{x})}, \ z_j = (x_j - \overline{x}) \tag{6}$$

where $z_i$ and $z_j$ are the standardization of the observed values in region $i$ and region $j$, respectively.

2.1.3. Spatial Econometric Model

The classical econometric method assumes that regions are independent. However, the flow of production factors can promote the improvement of AWGE in regions through the demonstration effect. Therefore, it is necessary to use the spatial econometric model to test this effect. Based on this, the present study used a spatial econometric model for empirical measurements.

The basic traditional spatial econometric models mainly include the spatial lag model (SLM), spatial error model (SEM), and spatial Durbin model (SDM). The SLM examines the spatial spillover effect caused by the spatial dependence of the variables, and the SEM examines the spillover effect of the impact of the error term in the adjacent areas on the regions [4]. Compared with the SLM and SEM models, the SDM model considers the spatial correlation of dependent variables as well as the spatial correlation of independent variables and has both spatial auto-correlation and spatial interaction effects. At the same time, for endogenous problems, the SDM model can be used to obtain estimated values that are not biased by amplification.

$$Y_{it} = \alpha_i + \rho \sum_{j=1}^{N} W_{ij} Y_{jt} + \beta X_{it} + \varphi \sum_{j=1}^{N} W_{ji} X_{jt} + U_i U_i = \lambda W \mu_i + \varepsilon_i \tag{7}$$

where $i$ and $j$ denote different regions; $W_{ij}$ denotes spatial weights; $X_{it}$ denotes explanatory variables; $Y_{it}$ denotes the AWGE of a region; $\beta$ is the regression coefficient of the explanatory variables; $\rho$ is the spatial regression coefficient of the explained variables; $\varphi$ is the spatial regression coefficient of the explanatory variables; and $\lambda$ is the spatial error regression coefficient.

If $\rho \neq 0, \varphi = 0$, then Equation (6) is a spatial lagged model (SLM). If $\lambda \neq 0, \rho = 0$, then Equation (6) is a spatial error model (SEM). If $\rho \neq 0, \varphi \neq 0, \lambda = 0$, then Equation (7) is a spatial Dubin model (SDM).

*2.2. Variable Selection*

2.2.1. AWGE as Explained Variable

The main purpose of the AWGE is to achieve the largest possible agricultural economy and grain output under a certain productivity level when the input of agricultural production factors such as water resources is fixed, gradually reducing the damage caused to the ecological environment by the pollutants generated in agricultural production [5,31]. This comprehensively reflects the coordinated development relationship between the agricultural economy, resource utilization, and environmental protection. This study took the narrow agriculture industry (planting industry) as the research object. Referring to relevant studies [5,32], we built an evaluation index system for AWGE, as shown in Table 1. Due to space limitations, relevant literature [5,31,32] can be referred to for specific calculation methods of agricultural non-point source pollution and agricultural carbon emissions. On this basis, MaxDEA software (MaxDEA Software Ltd., developed by Beijing Real World Research and Consultation Company Limited, Beijing, China) was used to calculate the AWGE in 87 regions in the Yellow River Basin from 2000 to 2019 based on the ratio of target water use to actual water use on the production frontier in the Super-SBM framework.

**Table 1.** Variable definition.

| Index | | Variable | Variable Definition |
|---|---|---|---|
| AWGE | Inputs | Land input | The total sown area of crops |
| | | Labor input | Agricultural employees are mainly employees in the primary industry × (total agricultural output value/total output value of agriculture, forestry, animal husbandry, and fishery) |
| | | Mechanical input | Total power of agricultural machinery |
| | | Water input | Agricultural irrigation water consumption; agricultural water is mainly used for irrigation |
| | | Chemical fertilizer input | The application amount of agricultural chemical fertilizer (net amount) |
| | | Pesticide input | The application amount of pesticides |
| | | Diesel input | The application amount of agricultural diesel |
| | | Plastic film input | The application amount of agricultural plastic film |
| | Desirable Outputs | Agricultural output | Total agricultural output, converted to constant price in 2000 |
| | | Grain output | Total grain output |
| | Undesirable Outputs | Agricultural carbon emissions | Direct or indirect carbon emissions from chemical fertilizers, pesticides, agricultural films, agricultural diesel, etc. |
| | | Agricultural non-point source pollution | Chemical fertilizer loss, pesticide residue, and agricultural film residue |

## 2.2.2. Explanatory Variable

To investigate the spatial effects of AWGE, the influence of natural endowment, technological progress, economic and social development, and other factors needed to be considered. AWGE in different regions is influenced by internal factors such as agricultural technical conditions, the popularization of mechanized services, and the popularization of water-saving technologies, which lead to changes in farmers' water use in agricultural production. Furthermore, the diversification of natural conditions, economic growth, urbanization, and other factors causes AWGE to change constantly [4,5]. Referring to existing studies [5,21–27] and considering the availability of data, variables influencing the AWGE were selected from the aspects of social economy, resource endowment, and technological progress. Specifically:

Water resource endowment: There is a negative effect of resource endowment on re-source utilization efficiency. The two water resources most directly related to the effect of regional AWGE are irrigation water and precipitation. In regions where water resources are relatively abundant, farmers may have poor awareness of water conservation. An unnecessary waste of water resources may occur in agricultural production, which increases the redundancy of agricultural water input and thus reduces AWGE. Referring to existing studies [4,21–23], per capita water resources, annual rainfall, and water structure were used to represent water resource endowment. Average annual rainfall (RAIN): data were from statistical yearbooks; per capita water resources (WATER): measured by the ratio of the total water resources of the region to the population of each region; water structure (WS): measured by the ratio of total agricultural water consumption to total regional water consumption. In order to reduce heteroscedasticity, this paper took the logarithm of per capita water resources.

Agricultural modernization level: Agricultural modernization is an effective way to realize the efficient development of agricultural water use. The degree of mechanization can represent the application degree of machinery in farming, irrigation, drainage, etc. The effective irrigation area refers to the area of land equipped with irrigation equipment capable of normal irrigation. Both of them are important indexes reflecting the development level of agricultural modernization. Improving the level of agricultural equipment can create an efficient agricultural production system, further enhance the comprehensive production capacity, and promote the improvement of AWGE. Referring to existing studies [4,23,24], the input intensity of agricultural machinery and effective irrigation level were used to represent the level of agricultural modernization. Input intensity of agricultural machinery (MA): measured by the ratio of the total power of agricultural machinery to the total sown

area of crops; effective irrigation level (GG): measured by the ratio of effective irrigation area to the total planting area of crops.

Economic and social development level: Economic and social development is the driving force to improve AWGE. Urbanization level and per capita GDP are important indicators to measure the level of economic and social development of a region. The higher the level of economic and social development is, the more farmers will be able to purchase and adopt efficient water-saving technologies and facilities, so as to improve the AWUE. The rural labor transfer has released rural surplus labor, realized the reconfiguration of family labor structure, and improved the AWGE. Referring to existing studies [5,21–27], urbanization, rural labor transfer, and rural per capita disposable income were adopted to represent the level of economic and social development. Rural labor transfer (RLT): the change in the rural labor force transferred from the agricultural sector to the non-agricultural sector; urbanization (URBAN): measured by the ratio of urban resident population to the total population; rural per capita disposable income (SR): data were from statistical yearbooks.

### 2.2.3. Data Sources

Referring to the research of Song et al. [33] and Li et al. [34], this paper took the natural basin of the Yellow River as the main body. In addition, considering the close economic relationship with the regions through which the Yellow River flows, the research scope was defined as the 87 regions above the prefecture level in the provinces where the Yellow River flows. At the same time, referring to the Yearbook of the Yellow River, the Yellow River Basin was divided into three regions: the upper, middle, and lower reaches. The period was 20 years, from 2000 to 2019. The data of the variables were derived from the China Rural Statistical Yearbook, the China Agricultural Statistical Report, the China Water Resources Bulletin, and the statistical yearbooks of 87 regions (Table 2). Some of the data were obtained from the local municipal governments according to public applications. The data of individual years were missing, and the interpolation method for adjacent years was used for smoothing. Table 3 shows the descriptive statistics of each variable.

**Table 2.** Date Sources.

| | Variable | Variable | Date Source |
|---|---|---|---|
| AWGE | Input | Land input/khm$^2$ | Statistical Yearbook of 87 regions |
| | | Labor input/$10^4$ people | China Rural Statistical Yearbook |
| | | Mechanical input/$10^4$ kW | Statistical Yearbook of 87 regions |
| | | Agricultural water input/$10^4$ m$^3$ | China Water Resources Bulletin |
| | | Chemical fertilizer input/$10^4$ t | Statistical Yearbook of 87 regions |
| | | Pesticide input/$10^4$ t | Statistical Yearbook of 87 regions |
| | | Diesel input/$10^4$ t | Statistical Yearbook of 87 regions |
| | | Plastic film input/$10^4$ t | Statistical Yearbook of 87 regions |
| | Desirable Output | Agricultural output/ Hundred million yuan | China Rural Statistical Yearbook and China Agricultural Statistical Report |
| | | Grain output/$10^4$ t | China Rural Statistical Yearbook and China Agricultural Statistical Report |
| | Undesirable Output | Agricultural carbon emissions/$10^4$ t | Statistical Yearbook of 87 regions |
| | | Agricultural non-point source pollution/$10^4$ t | Statistical Yearbook of 87 regions |
| Explanatory variable | WATER | Per capita water resources/m$^3$ | China Water Resources Bulletin |
| | RAIN | Annual rainfall/m$^3$ | China Water Resources Bulletin |
| | WS | Water structure/% | China Water Resources Bulletin |
| | URBAN | Urbanization/% | Statistical Yearbook of 87 regions |
| | SR | Rural per capita disposable income/Hundred million yuan | China Rural Statistical Yearbook and China Agricultural Statistical Report |
| | LT | Rural labor transfer/% | China Rural Statistical Yearbook and China Agricultural Statistical Report |
| | MA | Input intensity of agricultural machinery/% | China Rural Statistical Yearbook and China Agricultural Statistical Report |
| | GG | Effective irrigation level/% | Statistical Yearbook of 87 regions |

**Table 3.** Descriptive statistics of variables (The data of variables are from 87 regions in the Yellow River Basin from 2000 to 2019).

| | Variable | Variable | Mean | Std | Min | Max |
|---|---|---|---|---|---|---|
| AWGE | Input | Land input/khm$^2$ per year | 469.750 | 403.061 | 0.310 | 2011.980 |
| | | Labor input/$10^4$ people per year | 48.560 | 46.472 | 0.350 | 251.510 |
| | | Mechanical input/$10^4$ Kw per year | 359.870 | 347.168 | 3.100 | 1522.890 |
| | | Agricultural water input/$10^4$ m$^3$ per year | 3.810 | 6.967 | 0.001 | 48.850 |
| | | Chemical fertilizer input/$10^4$ t per year | 19.890 | 19.633 | 0.001 | 91.341 |
| | | Pesticide input/$10^4$ t per year | 0.430 | 0.517 | 0.001 | 2.191 |
| | | Diesel input/$10^4$ t per year | 6.100 | 6.392 | 0.003 | 35.912 |
| | | Plastic film input/$10^4$ t per year | 0.870 | 1.092 | 0.001 | 7.849 |
| | Desirable Output | Agricultural output/ Hundred million yuan per year | 134.540 | 129.261 | 0.660 | 585.280 |
| | | Grain output/$10^4$ t per year | 186.650 | 185.880 | 0.060 | 901.900 |
| | Undesirable Output | Agricultural carbon emissions/$10^4$ t per year | 28.620 | 25.723 | 0.001 | 116.250 |
| | | Agricultural non-point source pollution/ $10^4$ t per year | 13.230 | 13.001 | 0.001 | 60.550 |
| Explanatory variable | WATER | Per capita water resources/m$^3$ per year | 6.06 | 1.49 | 2.44 | 11.69 |
| | RAIN | Annual rainfall/m$^3$ per year | 0.61 | 0.19 | 0.16 | 1.36 |
| | WS | Water structure/% per year | 0.56 | 0.21 | 0.02 | 2.41 |
| | URBAN | Urbanization/% per year | 0.41 | 0.19 | 0.08 | 0.95 |
| | SR | Rural per capita disposable income/Hundred million yuan per year | 0.67 | 0.46 | 0.09 | 2.35 |
| | LT | Rural labor transfer/% per year | 0.23 | 0.08 | 0.01 | 0.78 |
| | MA | Input intensity of agricultural machinery/ % per year | 0.81 | 1.07 | 0.09 | 30.4 |
| | GG | Effective irrigation level/% per year | 0.38 | 0.18 | 0.02 | 0.95 |

## 3. Results

### 3.1. Calculation of AWGE in Yellow River Basin

Based on MAXDEA software and the adoption of the Super-SBM model, the AWGE of 87 regions in the Yellow River Basin from 2000 to 2019 was calculated (Table 4). The mean AWGE of the whole Yellow River Basin and the three regions in the upper, middle, and lower reaches were compared and analyzed (Figure 1).

**Table 4.** AWGE of the Yellow River Basin from 2000 to 2019.

| Upstream | 2000 | 2019 | Midstream | 2000 | 2019 | Downstream | 2000 | 2019 |
|---|---|---|---|---|---|---|---|---|
| Hohhot | 0.27 | 1.16 | Taiyuan | 0.21 | 0.46 | Jinan | 0.21 | 0.86 |
| Baotou | 0.12 | 1.08 | Datong | 0.23 | 1.00 | Qingdao | 0.41 | 1.06 |
| Wuhai | 0.06 | 1.19 | Yangquan | 0.66 | 1.00 | Zibo | 0.16 | 1.00 |
| Ordos | 0.13 | 1.00 | Changzhi | 0.77 | 0.49 | Zaozhuang | 0.37 | 1.00 |
| Bayan Nur | 0.05 | 1.00 | Jincheng | 1.00 | 0.68 | Dongying | 0.36 | 1.00 |
| Ulanqab | 1.00 | 0.76 | Shuozhou | 0.32 | 0.88 | Yantai | 0.38 | 1.00 |
| Alxa | 0.03 | 0.19 | Jinzhong | 0.16 | 0.48 | Weifang | 0.31 | 1.00 |
| Lanzhou | 0.15 | 0.38 | Yuncheng | 0.18 | 1.00 | Jining | 0.66 | 1.00 |
| Jiayuguan | 0.07 | 1.00 | Xinzhou | 0.34 | 1.00 | Tai'an | 0.32 | 0.68 |
| Jinchang | 0.10 | 0.42 | Linfen | 0.20 | 0.42 | Weihai | 1.00 | 1.00 |
| Baiyin | 0.15 | 0.33 | Luliang | 0.15 | 0.65 | Rizhao | 0.27 | 1.00 |
| Wuwei | 0.03 | 0.19 | Luoyang | 0.30 | 0.45 | Laiwu | 0.35 | 1.00 |
| Zhangye | 0.11 | 0.20 | Jiaozuo | 0.58 | 1.00 | Linyi | 0.39 | 1.00 |
| Jiuquan | 0.08 | 1.00 | Sanmenxia | 0.31 | 1.00 | Dezhou | 0.35 | 1.00 |
| Dingxi | 0.57 | 1.00 | Nanyang | 0.69 | 1.00 | Liaocheng | 0.35 | 0.92 |
| Linxia | 1.00 | 1.20 | Xi'an | 1.00 | 1.00 | Binzhou | 0.16 | 1.00 |

**Table 4.** *Cont.*

| Upstream | 2000 | 2019 | Midstream | 2000 | 2019 | Downstream | 2000 | 2019 |
|---|---|---|---|---|---|---|---|---|
| Gannan | 1.00 | 1.00 | Tongchuan | 1.00 | 1.00 | Heze | 0.33 | 1.00 |
| Xining | 0.19 | 0.35 | Baoji | 0.75 | 1.00 | Zhengzhou | 0.18 | 0.37 |
| Haidong | 0.17 | 0.42 | Xianyang | 0.37 | 1.00 | Kaifeng | 0.17 | 0.64 |
| Haibei | 0.07 | 0.21 | Weinan | 0.21 | 0.34 | Pindingshan | 0.37 | 0.82 |
| Huangnan | 0.53 | 1.00 | Yan'an | 1.00 | 1.00 | Anyang | 0.22 | 0.60 |
| Hainan | 0.16 | 0.19 | Yulin | 0.28 | 0.99 | Hebi | 0.35 | 1.00 |
| Guoluo | 1.02 | 1.00 | Ankang | 1.00 | 1.00 | Xinxiang | 1.00 | 1.00 |
| Yushu | 1.00 | 1.00 | Shangluo | 1.16 | 1.00 | Puyang | 0.20 | 0.60 |
| Haixi | 0.04 | 0.07 | Tianshui | 0.33 | 1.00 | Xuchang | 1.00 | 1.00 |
| Yinchuan | 0.06 | 0.08 | Pingliang | 0.77 | 0.95 | Luohe | 0.28 | 1.09 |
| Shizuishan | 0.06 | 0.10 | Qingyang | 1.00 | 1.00 | Shangqiu | 0.65 | 1.00 |
| Wuzhong | 1.00 | 1.00 | | | | Xinyang | 0.67 | 1.00 |
| Guyuan | 0.31 | 1.20 | | | | Zhoukou | 0.98 | 1.00 |
| Zhongwei | 0.06 | 0.08 | | | | Zhumadian | 0.61 | 1.00 |

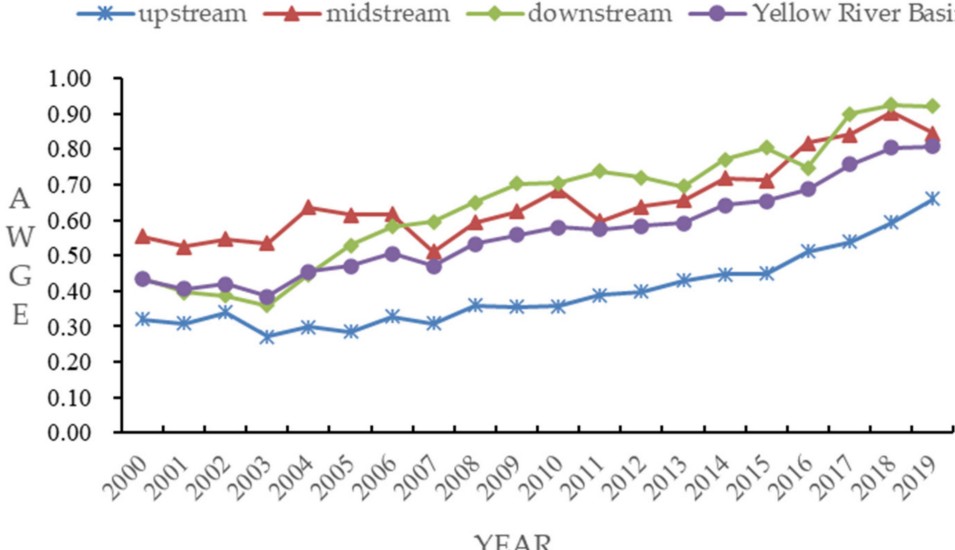

**Figure 1.** Evolution trend of AWGE in the Yellow River Basin from 2000 to 2019.

By observing the trends in Figure 1, the AWGE in the Yellow River basin generally shows a trend of slow improvement, but the average value of each year is below 0.9, and there is still room for improvement. From 2000 to 2019, the AWGE showed a stable trend, with a small range of change, and the overall AWGE was at a low level. After 2011, the AWGE of the Yellow River Basin showed a steady upward trend. The possible reason is that since the Chinese government explicitly required the implementation of the strictest water resource management system to date in 2011 and the Comprehensive Plan for the Yellow River Basin in 2013, a number of measures have been taken, such as the rational allocation and efficient utilization of water resources, and water ecological protection to promote water conservation and efficiency, which has led to a significant increase in AWGE in the Yellow River Basin.

By comparing the calculation results of the three regions in the upper, middle, and lower reaches, the regions with high AWGE are concentrated in the lower reaches, and the regions with low AWGE are located upstream, which is basically the same as the pattern of economic level difference. With the continuous development of the economy, the agricultural technology level in the downstream region has made remarkable progress, and more attention has been paid to agricultural modernization and large-scale development, consciously improving the coordination between agricultural production, resource conser-

vation, and environmental protection. Compared with the downstream region, the overall economic and social development level of the upstream region is relatively low, with a slow transformation of agricultural modernization and an extensive agricultural development mode. Therefore, the enhancement of AWGE is relatively slow. It is worth noting that the midstream area of Shanxi, the Shaanxi gorge region, due to its rich resources and good hydropower development conditions, showed a high level of AWGE in the early stage but was later surpassed by the downstream region. However, with the implementation of Western Development, the Belt and Road Initiative, and other policies, the midstream region represented by Shaanxi Province has enjoyed the benefits of national policies, and the AWGE has been greatly improved, which has gradually narrowed the average gap between the midstream and the upstream regions.

Spatial Pattern and Differentiation Characteristics of AWGE in the Yellow River Basin

In order to further reveal the spatial aggregation changes and differentiation characteristics of AWGE in the Yellow River Basin, the spatial association and dependence characteristics of AWGE were analyzed using the global Moran I and LISA aggregation analysis methods in the ESDA series, supported by ArcGIS and the GeoDa method. Table 5 reports the global Moran I of AWGE in the Yellow River Basin from 2000 to 2019. It can be found that the interval range of Moran's I of AWGE characterized by AWGE over the years is [0.261, 0.373], all of which are a positive pass of the test of the 5% significance level, indicating that the AWGE in the Yellow River Basin presents a significant spatial auto-correlation. That is, the AWGE at the prefecture-level city level has positive clustering and dependence characteristics. The geographical spatial pattern is an important factor affecting the AWGE in the Yellow River Basin. There is a spatial dependence on the AWGE among neighboring regions. The AWGE in a city will not only affect neighboring regions but may also be affected by the AWGE in neighboring regions.

**Table 5.** Moran's I index.

| Year | Moran | Z | $p$ Value | Year | Moran | Z | $p$ Value |
|------|-------|-------|-----------|------|-------|-------|-----------|
| 2000 | 0.272 | 4.053 | 0.000 | 2010 | 0.275 | 4.081 | 0.000 |
| 2001 | 0.283 | 4.222 | 0.000 | 2011 | 0.242 | 3.612 | 0.000 |
| 2002 | 0.291 | 4.327 | 0.000 | 2012 | 0.206 | 3.110 | 0.001 |
| 2003 | 0.330 | 4.905 | 0.000 | 2013 | 0.171 | 2.617 | 0.004 |
| 2004 | 0.387 | 5.699 | 0.000 | 2014 | 0.203 | 3.066 | 0.001 |
| 2005 | 0.337 | 4.973 | 0.000 | 2015 | 0.244 | 3.647 | 0.000 |
| 2006 | 0.302 | 4.474 | 0.000 | 2016 | 0.145 | 2.245 | 0.012 |
| 2007 | 0.284 | 4.220 | 0.000 | 2017 | 0.266 | 3.979 | 0.000 |
| 2008 | 0.246 | 3.672 | 0.000 | 2018 | 0.311 | 4.634 | 0.000 |
| 2009 | 0.266 | 3.953 | 0.000 | 2019 | 0.335 | 4.982 | 0.000 |

Next, according to the significant spatial correlation between the AWGE among prefecture-level regions, under the 95% confidence interval, the AWGE is further divided into four types with different levels: (1) High–High Cluster (H–H), meaning the AWGE of a city and its neighboring regions is high. (2) Low–Low Cluster (L–L), meaning the AWGE in a specific area and in adjacent areas is low. (3) High–Low Outlier (H–L), meaning the AWGE in one area is high, but the AWGE in neighboring areas is low. (4) Low–High Outlier (L–H), meaning the AWGE in one area is low, but that of its neighboring regions is higher. In this part, six typical time points in 2000, 2004, 2008, 2012, 2016, and 2019 are selected for LISA clustering analysis (Figure 2).

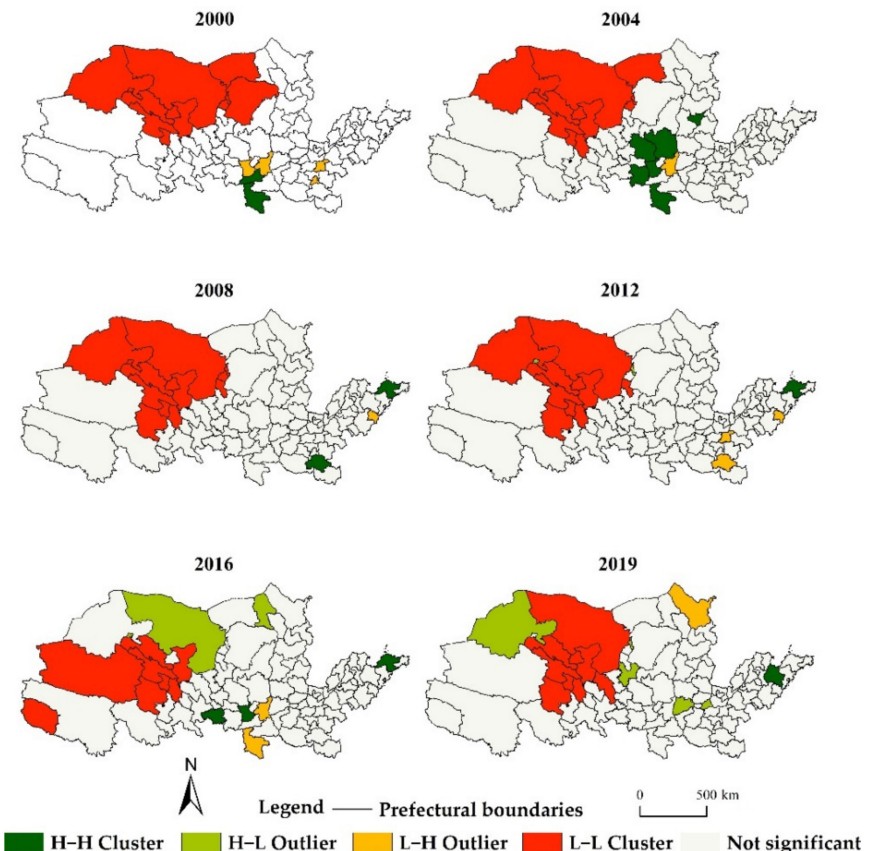

**Figure 2.** The LISA aggregation of AWGE in the Yellow River Basin.

The spatial agglomeration characteristics of AWGE in the Yellow River Basin can be clearly seen from the LISA agglomeration map. The spatial agglomeration pattern of AW-GE in general remained basically stable; specifically:

(1) The H–H clusters of AWGE in the Yellow River Basin were scattered during the investigation period. Before 2008, H–H was mainly located in Shaanxi Province in the midstream area of the Yellow River Basin. The natural resource endowment in the region was strong, so the AWGE was at a high level. Since 2008, the H-H shifted from east to west and from north to south, and the scope of H–H gradually narrowed down in northern Shandong. The agricultural water-saving technology in these regions is relatively developed, and the AWGE in specific areas and in neighboring regions is relatively high, which had a significant positive impact on the neighboring regions. However, the levels of AWGE in other areas and in adjacent areas were also high, and the high-level clustering feature is not obvious, which indirectly reflects that the gap between the AWGE in the midstream and downstream is narrowing.

(2) The L–L of AWGE in the Yellow River Basin from 2000 to 2019 was more frequent and more stable and was basically located in the northern part of the upstream region, such as in Gansu and Qinghai. The upstream region was already facing multiple tests, such as a fragile ecological environment and poor ecological carrying capacity. Although Gansu and Qinghai have rich resource reserves, the extensive agricultural production and relatively backward agricultural technology resulted in the AWGE being at a low level for a long time, forming an inefficient agricultural water use mode with the surrounding low–efficiency areas.

(3) During the study period, the H–L and L–H were fewer and sporadic. Qinghai in the west of the upstream was mainly L–H. The AWGE in these areas was low, and the agricultural production foundation was weak, but the AWGE of neighboring regions was relatively high. H–L was mainly distributed in the border area between H–H and

L–L. From 2000 to 2019, the range of H–L gradually increased, while the range of L–H gradually decreased. The AWGE in all regions was basically gradually improved.

### 3.2. Spatial Effect Analysis of AWGE in the Yellow River Basin

3.2.1. Spatial Correlation Test and Model Selection

(1) The premise of building a spatial econometric model is to set the spatial weight matrix. In this paper, a rook adjacency weight matrix W1 with a common boundary was constructed. When two regions had a common boundary, the element in the matrix is set to 1; otherwise, it was set to 0.

(2) Moran's I was used to test the spatial correlation of AWGE. The above test results (Table 5) show that the Moran's index of the AWGE under the adjacent W1 is significantly positive [0.261, 0.373], indicating that the AWGE among regions shows a significant positive spatial auto-correlation.

(3) The commonly used spatial econometric models mainly include the spatial lag model (SLM), spatial error model (SEM), and spatial Dobbin model (SDM). The progressiveness LM Test selects among the three models by conducting the Wald test and LR test. Generally speaking, when only one of LM-lag and LM-error passes the significance test, it will be directly selected between SLM and SEM models. If both pass the significance test, further LR and Wald tests will be conducted for final judgment. Table 6 reports the results of spatial diagnosis.

**Table 6.** Spatial correlation test results.

| Test Statistic | $\chi^2$ | $p$ |
| --- | --- | --- |
| LM-lag | 83.761 | 0.000 |
| LM-error | 74.530 | 0.000 |
| Wald-spatial lag | 16.000 | 0.042 |
| Wald-spatial error | 16.390 | 0.037 |
| LR-spatial lag | 15.940 | 0.043 |
| LR-spatial error | 16.450 | 0.036 |
| Hausman test | 126.540 | 0.000 |
| Spatial fixed effect LR-test | 38.180 | 0.000 |
| Time fixed effect LR-test | 1484.620 | 0.000 |

It can be seen from Table 6 that the LM value of the model passed the test at a significance level of 5%, which indicates that the model has the characteristics of an SLM model and SEM model at the same time. In the case that neither of the two models can be rejected, the LR and Wald tests were further conducted in this paper, and the test results were still significant. Therefore, it is determined that the SDM model cannot degenerate into an SLM model or SEM model, and the SDM model can best fit the data. At the same time, according to the joint significance and Hausman test results, the spatial double fixed effect SDM model was finally selected for spatial effect analysis.

3.2.2. Analysis of Benchmark Regression Results

Table 7 reports the model estimation results. For comparison and analysis, it also reports the results of the OLS model and spatial econometric models (SEM, SAR, and SDM). From the regression estimation results, the significance and direction of the influence of various factors on the AWGE in different models are generally the same, showing good robustness. Meanwhile, the coefficients of the spatial spillover effect are significantly positive, indicating that there is a significant spatial spillover effect in the AWGE among regions, which means that the AWGE in this region will have a strong demonstration effect and radiation-driving effect on the AWGE in adjacent regions.

**Table 7.** Benchmark regression results.

| Variable | OLS | Spatial Econometric Model | | |
| --- | --- | --- | --- | --- |
| | | **SEM** | **SAR** | **SDM** |
| RAIN | 0.108 * | 0.131 | 0.125 | 0.182 |
| | (0.057) | (0.083) | (0.078) | (0.132) |
| SR | 0.268 *** | 0.279 *** | 0.262 *** | 0.231 *** |
| | (0.018) | (0.039) | (0.037) | (0.051) |
| LT | 0.126 | −0.205 | −0.187 | −0.277 * |
| | (0.148) | (0.163) | (0.160) | (0.166) |
| LnWATER | −0.003 | −0.007 | −0.007 | −0.016 |
| | (0.010) | (0.014) | (0.013) | (0.015) |
| URBAN | −0.003 | 0.182 ** | 0.171 ** | 0.120 |
| | (0.011) | (0.080) | (0.078) | (0.088) |
| MA | −0.015 *** | −0.015 *** | −0.015 *** | −0.014 *** |
| | (0.005) | (0.005) | (0.004) | (0.005) |
| GG | −0.473 *** | 0.216 *** | 0.222 *** | 0.164 ** |
| | (0.044 | (0.072) | (0.071) | (0.074) |
| WS | −0.345 *** | −0.295 *** | −0.291 *** | −0.309 *** |
| | (0.047) | (0.049) | (0.048) | (0.049) |
| W × Rain | | | | −0.205 |
| | | | | (0.172) |
| W × SR | | | | 0.0666 |
| | | | | (0.074) |
| W × LT | | | | 0.524 * |
| | | | | (0.298) |
| W × LnWATER | | | | 0.041 |
| | | | | (0.025) |
| W × URBAN | | | | 0.048 |
| | | | | (0.133) |
| W × MA | | | | 0.014 |
| | | | | (0.012) |
| W × GG | | | | 0.275 * |
| | | | | (0.147) |
| W × WS | | | | 0.176 * |
| | | | | (0.100) |
| $\rho$ | | 0.118 *** | 0.115 *** | 0.106 *** |
| | | (0.034) | (0.033) | (0.034) |
| Sigma$^2$ | | 0.027 *** | 0.027 *** | 0.026 *** |
| | | (0.001) | (0.001) | (0.001) |
| C | 0.498 *** | | | |
| | (0.083) | | | |
| R2 | 0.374 | 0.381 | 0.381 | 0.386 |
| LogL | | 667.538 | 667.795 | 675.766 |
| Observation | 1740 | 1740 | 1740 | 1740 |

Note: ***, **, and * are significant at 1%, 5%, and 10%, respectively. Robust standard error in parentheses.

The coefficient of SR is positive and passed the significance test at a level of 1%, implying that SR has a significant positive influence on the AWGE. In areas with a high level of economic development, farmers are more likely to accept advanced agricultural technology concepts and have the ability to purchase and adopt efficient water-saving technologies and facilities in agricultural production, thus contributing to the improvement of AWGE. The coefficient of GG is positive and passed the significance test at a level of 5%, which means that the GG has a significant positive influence on the AWGE. Improving the GG can improve the extensive irrigation mode in the Yellow River Basin, reduce the amount of agricultural water use, and reduce the redundant input of water resources, which will improve the AWGE.

The coefficient of LT is negative and passed the significance test at a level of 10%, which means that LT has a significant negative influence on the AWGE. The possible reason is that

with the rapid development of urbanization, non-agricultural employment opportunities are also increasing, and the personnel left behind in villages reduce the actual labor input in agricultural production, which has a negative influence on the AWGE. The coefficient of MA is negative and passed the significance test at a level of 1%, which signifies that MA has a significant negative influence on AWGE. Behind this possibly lies the reason that the input of agricultural machinery increases agricultural non-point source pollution and carbon emissions through diesel consumption and agricultural film coverage. The superposition of these factors will not be conducive to the improvement of AWGE. The coefficient of WS is negative and passed the significance test at a level of 1%, meaning that a higher proportion of agricultural water in the total water consumption will have a significant negative influence on the AWGE. One possible reason for this is that the lack of awareness of water conservation among farmers results in the redundancy of agricultural water inputs, which leads to the significant negative influence of the WS on the AWGE.

### 3.2.3. Analysis of Regional Heterogeneity Results

Generally speaking, significant systematic differences exist in the level of economic and social development, agricultural production, resource endowment, and other aspects in various regions, and their influence on the AWGE may also be distinct. Therefore, it is necessary to consider regional heterogeneity in the measurement model. In order to further investigate the regional heterogeneity of the influence of various factors on the AWGE, a sub-sample investigation was conducted in the upstream, midstream, and downstream regions of the Yellow River Basin. The SDM model was still used as the test model for spatial econometric analysis. Table 8 reports the estimation results of the upstream, midstream, and downstream regions of the Yellow River Basin.

**Table 8.** Regression results in different regions.

| Variable | Upstream | Midstream | Downstream |
|---|---|---|---|
| RAIN | −0.650 *** (0.251) | 0.367 (0.257) | 0.395 (0.152) |
| SR | 0.596 *** (0.075) | −0.037 (0.096) | 0.125 (0.090) |
| LT | −0.429 ** (0.215) | −0.365 (0.583) | 0.883 *** (0.284) |
| LnWATER | −0.023 (0.021) | −0.033 (0.032) | −0.001 (0.023) |
| URBAN | −0.060 (0.135) | 0.298 * (0.164) | 0.334 ** (0.154) |
| MA | −0.014 *** (0.005) | −0.068 (0.058) | −0.169 *** (0.063) |
| GG | 0.255 *** (0.082) | 0.545 ** (0.242) | −0.889 *** (0.244) |
| WS | −0.529 *** (0.105) | −0.144 ** (0.068) | −0.388 *** (0.081) |
| W × Rain | 0.828 ** (0.411) | −0.185 (0.386) | −0.556 *** (0.239) |
| W × SR | −0.536 *** (0.115) | −0.077 (0.184) | 0.659 *** (0.141) |
| W × LT | 1.642 *** (0.438) | 1.222 (1.123) | 1.539 ** (0.531) |
| W × LnWATER | −0.037 (0.045) | −0.008 (0.064) | 0.052 (0.039) |
| W × URBAN | −0.260 (0.227) | 0.166 (0.339) | 0.350 (0.258) |

**Table 8.** *Cont.*

| Variable | Upstream | Midstream | Downstream |
|---|---|---|---|
| W × MA | −0.011 | 0.197 * | −0.419 *** |
| | (0.014) | (0.108) | (0.116) |
| W × GG | 0.934 *** | 0.534 | −0.054 |
| | (0.171) | (0.488) | (0.494) |
| W × WS | −0.035 | 0.049 | 0.423 *** |
| | (0.208) | (0.115) | (0.146) |
| $\rho$ | −0.144 ** | 0.106 * | −0.005 |
| | (0.064) | (0.059) | (0.052) |
| Sigma$^2$ | 0.025 *** | 0.026 *** | 0.015 *** |
| | (0.001) | (0.001) | (0.001) |
| R2 | 0.099 | 0.183 | 0.528 |
| LogL | 252.232 | 210.755 | 389.219 |
| Observation | 600 | 540 | 600 |

Note: ***, **, and * are significant at 1%, 5%, and 10%, respectively. Robust standard error in parentheses.

The results show that the spatial spillover effect is still significant, but the influence of various factors on the AWGE in the upstream, midstream, and downstream regions is diverse in degree and significance. Specifically:

There is a significant positive influence of SR and GG on AWGE in upstream regions, a significant positive influence of URBAN and GG on AWGE in midstream regions, and a significant positive influence of LT and URBAN on AWGE in downstream regions. The upstream, midstream, and downstream regions represent different stages of agricultural development. The economic development of the upstream region is still at a low level compared with other regions. Therefore, economic development factors still play an important role in the development process of improving the AWGE in upstream regions. The increase in GG can improve irrigation practices in the upstream and midstream regions of the Yellow River Basin, reduce agricultural water use, and reduce water input redundancy, which, in turn, improves the AWGE.

Due to the better social–economic development foundation and relatively perfect agricultural production technology in the midstream and downstream regions, the AWGE has generally reached a high level. It is of little developmental potential to rely on the input of traditional agricultural factors to improve the AWGE. However, the positive effects of scale effect, cost saving, knowledge, and technology spillover brought by economic agglomeration generated by urbanization development are conducive to the development of agricultural production equipment, water-saving technology, and water conservancy facilities in downstream regions, thus contributing to the improvement of AWGE. At the same time, the rapid development of urbanization in the downstream region has further attracted the labor force. The cross-regional flow and transfer of the rural labor force between adjacent regions can not only change the input structure of factors in the areas where the labor force is transferred out, but also drive agricultural marketization, raise the agricultural technology level, and further promote the AWGE in downstream regions.

## 4. Conclusions

At present, China's economy has entered a new stage, in which economic development is gradually transforming from an extensive mode of pursuing growth rate to a green development mode of seeking structural adjustment and environmental efficiency. In this paper, the Super-SBM model was used to measure the AWGE in 87 regions in the Yellow River Basin from 2000 to 2019. Through the global Moran's I index and LISA agglomeration map, the spatial and temporal dynamic evolution and differentiation characteristics of AWGE were discussed, and then a spatial econometric model was constructed to analyze the factors influencing AWGE. The following conclusions are drawn:

From the trend of time evolution, the AWGE in the Yellow River Basin shows a steady upward trend, but there is still a certain distance from the effective front, and there is

still room for improvement in the AWGE. From the perspective of regions, there exists an obvious gradient difference in AWGE. The midstream and downstream regions are at a higher level, and the discrepancy between them is lessening year by year with the upstream reaches at a lower level.

From the spatial evolution pattern, the AWGE in the Yellow River Basin shows significant, positive clustering and dependence characteristics in terms of spatial distribution. On the whole, it presents a spatial pattern of L–L clusters mainly in Gansu and Qinghai in the upper reaches; H–H clusters mainly in Shaanxi, Shandong, and Henan in the midstream and downstream regions; and H–L and L–H outliers in the periphery. The distribution pattern is basically stable.

The change in AWGE is influenced by many factors, such as resource endowment, social and economic development, and agricultural modernization development, and there is a significant spatial spillover effect. From the perspective of the Yellow River Basin as a whole, SR and GG have a significant positive influence on the AWGE, while LT, MA, and WS have a significant negative influence on the AWGE. From the sub-basin results, the spatial spillover effect remains significant. Additionally, the influence of each factor on the AWGE in the upstream, midstream, and downstream region differ in degree and significance.

Based on the above analysis, some policy implications for improving AWGE in the Yellow River Basin can be made:

First, in accordance with the inherent requirements for the development of agricultural water conservation and efficiency in the National Water Conservation Action Plan, each local government should pay attention to the differences between regions and formulate local policies according to local conditions to promote the improvement of AWGE.

Second, full attention should be paid to the spatial effects of AWGE. The restrictive factors affecting the reallocation of agricultural production factors between regions should be reduced by breaking down administrative hierarchical barriers and local protectionism. Good interaction should be formed and should cycle between regions to achieve the goal of balanced AWGE development.

Third, the functional positioning of each region under the synergistic development strategy is clarified. For the upstream region, combined with its underdeveloped economy and crude agricultural production methods, there is an urgent need to accelerate the transformation of traditional agriculture to modern agriculture and to further improve farmers' income levels. For the midstream region, it should continue to promote the promotion of effective irrigation areas and to improve the effective utilization coefficient of irrigation water in farmland. For the downstream region, the agglomeration effect of urbanization should be actively brought into play, and relevant technologies should be promoted and demonstrated to form leading and demonstration areas for water saving and efficiency enhancement in the basin.

This study enriches the content surrounding the evaluation and spatial effect of AWGE in the Yellow River Basin and provides a new way of thinking and a new theoretical reference for the high-quality development of water resources and food security in the Yellow River Basin. Because the planting structure of grain crops in different regions is quite different, the characteristics of water demand, water consumption, yield, and value of different crops are different. Because the research focus of this paper is on the water use efficiency evaluation index related to water resource utilization and output and considers the limitation of content space, the water use efficiency of various crops or major crops in different regions is not presented. Therefore, how to reflect the AWGE of different crops and conduct a comparative analysis is also a direction for improvement in future research.

**Author Contributions:** Data curation, R.D. and S.C.; funding acquisition, W.L. (Weinan Lu); methodology, W.L. (Wenxin Liu); resources, X.G. and B.Z.; writing—review and editing, W.L. (Weinan Lu) and B.Z. All authors have read and agreed to the published version of the manuscript.

**Funding:** This research was funded by the Social Science Major Project of Shaanxi Province: 2022ND0280, 2021ND0378; the Social Science Foundation of the Ministry of Education: 21YJC630086; the Research Funds of Northwest A & F University (Z1090220194); and the Natural Science Basic Research Program of Shaanxi: 2020JQ-282.

**Institutional Review Board Statement:** This study mainly focused on models and data analysis and did not involve human factors considered dangerous. Therefore, ethical review and approval were waived for this study.

**Informed Consent Statement:** Not applicable.

**Data Availability Statement:** Data are available on request due to restrictions, e.g., privacy or ethical. The data presented in this study are available on request from the corresponding author. The data are not publicly available due to the strict management of various data and technical resources within the research teams.

**Acknowledgments:** We would like to thank the American journal experts who edited this paper. We also appreciate the constructive suggestions and comments on the manuscript from the reviewer(s) and editor(s).

**Conflicts of Interest:** The authors declare no conflict of interest.

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
