# Peer review of "Spatial–Temporal Dynamic Evolution and Influencing Factors of Green Efficiency of Agricultural Water Use in the Yellow River Basin, China"

_water, doi:10.3390/w15010143_

Round 1

Reviewer 1 Report

This paper covers a topic about water use efficiency. In general, the topic is not new and I did not get any new information from the study. In addition, the paper is not well written and not easy to follow. Thus, the current quality of the paper is not suitable for publication.

 Title

The title is not clear and concise. There are two “spatial” in the title, which looks redundant. What is “agricultural water use green efficiency”? Is it “green water use efficiency”?

 Abstract

Abstract is sufficiently informative especially when read in isolation along with the appropriate key words. However, I feel the authors can modify the abstract and focus more on the key results obtained, conclusion what they are arrived, and the political implications of the study rather than the general remarks about the work. In short words, precisely they need to come out with the recommendation statement. From the abstract, I did not see any implication of the study? Why did you conduct this study? What’s the scientific question?

Please define Super-SBM when it first appears in the abstract. The language should be improve.

Key words

The current key words are highly repetitive with the concept including in the topic.

 Introduction/Materials and Methods/Results/Conclusions/References

These sections are not well-organized, not easy to follow and should be greatly improved.

Author Response

Response to Reviewer 1 Comments

Point 1: This paper covers a topic about water use efficiency. In general, the topic is not new and I did not get any new information from the study. In addition, the paper is not well written and not easy to follow. Thus, the current quality of the paper is not suitable for publication. The paper needs careful language, spellings, redundant words and sentence corrections in the entire document. Please see the suggestions highlighted in the comments box. 

Response 1: Thank you for this helpful suggestion. This manuscript has been revised carefully. I have corrected the problems that the reviewers brought forward point by point and have replied carefully, addressing the problems raised. Indeed, the topic is not very new. So we have tried to innovate in terms of research content and research methods.

About the study area, the current paper takes the Yellow River Basin as an example. The Yellow River, the second longest river in China and the mother river of China, supports 9% of China's population and 15% of China's arable land with about 2% of its  runoff. As an important grain base and ecological functional area in China, the status of agricultural water resources in the Yellow River Basin is related to China's grain and ecological security. However, with the rapid economic development and population growth in the Yellow River Basin, the scale of water resources development and utilization along the Yellow River has been expanding, leading to increasingly serious problems such as the out-standing contradiction between the supply and demand of its agricultural water resources and fragile ecological environment. In particular, it poses a great threat to agricultural activities within the Yellow River Basin, which is mainly irrigated. Therefore, it represents an important theoretical reference and has practical significance for agricultural water saving and efficiency enhancement, food security protection, and ecological environment improvement in the Yellow River Basin, enabling it to be taken as the research area, to explore the spatial–temporal evolution characteristics of AWGE and identifying the key factors influencing the improvement in AWGE.

About the methods, the scientific evaluation of agricultural water use effiency (AWUE) is a basic prerequisite for the overall improvement of AWUE. In general, the evaluation of AWUE is a multi-objective and multi-criterion synthetical problem in essence, related evaluation studies have been developed and refined in the course of production and life, resulting in different types of AWUE. According to its development history, there has been a gradual shift from the study of engineering efficiency of irrigation water delivery and field utilization to various efficiency studies with water productivity as an indicator. Hu et al. measuring measured water use efficiency (WUE) in the framework of total factor production by the “ratio of target water consumption to actual quantity”. This measurement idea considers the contribution of various input factors to on economic growth, so as to measure the macro-comprehensive economic benefits of a resource system more truly and objectively. Since then, the total factor water use efficiency (TWUE) has gradually been recognized and applied by the academic community. On the basis of the this evaluation, scholars have measured the AWUE by stochastic frontier function method (SFA), data envelopment analysis (DEA), and other methods.

Generally, the existing literatures have has laid a firm foundation for the indepth study of AWUE, existing studies only measure the AWUE in a region using the efficiency value of a single factor or by only considering the desirable output while neglecting the agricultural non-point source pollution and other undesirable outputs due to negative externalities during the process of agricultural economic growth in the assessment. Secondly, existing studies often ignore the impact of geographical and spatial factors on the development of regional AWUE when using the traditional panel model for analysis. However, in the process of agricultural production, due to the similarity of natural resource endowment, economic development level, and agricultural water use mode in adjacent areas, the AWUE between different regions may have mutual spatial influence. Thirdly, fewer studies have been conducted on AWUE in Yellow River Basin, and the limited research is only from at the a national level to and analyzes the AWUE differences between provinces in Yellow River Basin. However, the important role of local and municipal governments in coordinating the allocation of resources within cities makes the study of mesogeographic units necessary. Thus, the study at the local and municipal levels allows for a more in-depth examination of the characteristics and unevenness of AWUE development between regions.

Based on this, the current paper takes the Yellow River Basin as an example, taking agricultural carbon emissions and agricultural non-point source pollution as undesirable output indicators, the Super-SBM model is used to measure the AWGE of 87 cities in the Yellow River Basin from 2000 to 2019. Through the global Moran's I index and LISA agglomeration map, the spatial and temporal dynamic evolution and differentiation characteristics of AWGE are discussed, and then a spatial econometric model was constructed to analyze the factors influencing AWGE. Finally, according to the qualitative analysis and empirical tests summed yield up the relevant policy implications.

Point 2: The title is not clear and concise. There are two “spatial” in the title, which looks redundant.

Response 2: Thank you for this helpful suggestion. We thank your comments which helped us improve the quality of this manuscript. According to your advice, we have created a new title, “Spatial–Temporal Dynamic Evolution and Influencing Factors of Agricultural Water Use Green Efficiency in the Yellow River Basin, China”. 

Point 3: What is “agricultural water use green efficiency”? Is it “green water use efficiency”?

Response 3: Thank you for this helpful suggestion. We thank your comments which helped us improve the quality of this manuscript.

In general, the evaluation of agricultural water use efficiency (AWUE) is a multi-objective and multi-criterion synthetical problem in essence, related evaluation studies have been developed and refined in the course of production and life, resulting in different types of AWUE. According to its development history, there are three specific categories, which are distinguished mainly by differences in output. The early AWUE studies simply considered the economic benefits brought by agricultural water use, i.e., the total agricultural output value as the only output, and the maximum economic benefits with the minimum water input. Therefore, the traditional AWUE is also called agricultural water use economic efficiency. However, water resources bring undesired outputs in addition to economic benefits in the agricultural production process. Therefore, the inclusion of undesired outputs in the study of AWUE is more in line with the actual production process of agricultural development, that is, the agricultural water use environmental efficiency. With the concept of "Green Development", the study of AWUE, which only considers economic and environmental benefits, no longer meets the requirements of today's agricultural development. The main purpose of the agricultural water use green efficiency (AWGE) is to achieve the largest possible agricultural economy and grain output under a certain productivity level when the input of agricultural production factors such as water resources is fixed, gradually reducing the damage caused by the pollutants generated in agricultural production to the ecological environment. This comprehensively reflects the coordinated development relationship among agricultural economy, resource utilization, and environmental protection, which is the core idea of AWGE. The agricultural water use economic efficiency, the agricultural water use environmental efficiency and the agricultural water use green efficiency are both different and related. AWGE is the enrichment and improvement of economic efficiency and environmental efficiency, so that the evaluation system of AWUE tends to be improved continuously.

Point 4: Abstract is sufficiently informative especially when read in isolation along with the appropriate key words. However, I feel the authors can modify the abstract and focus more on the key results obtained, conclusion what they are arrived, and the political implications of the study rather than the general remarks about the work. In short words, precisely they need to come out with the recommendation statement. From the abstract, I did not see any implication of the study? Why did you conduct this study? What’s the scientific question?

Response 4: We appreciate the reviewer’s comment. We have edited the Abstract to improve clarity. It is that “The progression of agricultural production, ever-increasing industrialization, population boom, and more water-concentrated lifestyles had a severe burden on Yellow River Basin’s existing water resources, particularly in the current century. In the context of resource and environmental constraints, improving agricultural water use green efficiency (AWGE) is an important measure to for alleviatinge the shortage of water resources as well as meeting, the intrinsic requirement to promote the green transformation and upgrading of agriculture. This study used the Super Slack-Based Measure (Super-SBM) to measure the AWGE of 87 cities in the Yellow River Basin from 2000 to 2019. Based on spatial and temporal perspectives, it applied Exploratory Spatial Data Analysis (ESDA) to explore the dynamic evolution and regional differences in AWGE. Then, this study used a spatial econometric model to analyze the main factors that influence AWGE in the Yellow River Basin. The results show that firstly, the AWGE of the Yellow River Basin shows a steady upward trend from 2000 to 2019, but the differences among cities were obvious. Secondly, the AWGE showed an obvious spatial autocorrelation in the Yellow River Basin and showed significant high–high and low–low agglomeration characteristics. Thirdly, rural per capita disposable income and effective irrigation have positive influence on AWGE, while rural labor transfer, input intensity of agricultural machinery and water structure have negative influence. The spatial econometric model regression results show that the influence factors of AWGE in the Yellow River Basin showed significant spatial spillover effects and spatial heterogeneity in their effect. Finally, when improving AWGE in the Yellow River Basin, plans should be formulated according to local conditions. The results of this study can provide new ideas on the study of AWGE in the Yellow River Basin and provide references for the formulation of regional agricultural water resource utilization policies as well.”(Line 14-35)

Point 5: Please define Super-SBM when it first appears in the abstract.

Response 5: Thank you for this helpful suggestion. We thank your comments which helped us improve the quality of this manuscript. Super-SBM is used to define Super Slack-Based Measure. We have defined this in the abstract section.  (Line 63)

Point 6: The language should be improve.

Response 6: Thank you for this helpful suggestion. We thank your comments which helped us improve the quality of this manuscript. For some of the more verbose sentence and phrases to cut to make them more fluent to read. By re-organizing, we have made the language more concise and easier to understand. At the same time, I have used English editing from MDPI to revised it. It is that english-editied-53445.

Point 7: The current key words are highly repetitive with the concept including in the topic.

Response 7: Thank you for this helpful suggestion. We thank your comments which helped us improve the quality of this manuscript. We have changed the keyword to “agricultural water use; Yellow River Basin; super-SBM model; spatial econometric model;

Point 8: These sections are not well-organized, not easy to follow and should be greatly improved.

Response 8: Thank you for this helpful suggestion. This manuscript has been revised carefully. I have corrected the problems that the reviewers brought forward point by point and have replied carefully, addressing the problems raised. And all revisions are clearly highlighted by using the "Track Changes" function in Microsoft Word 2016, so that changes are easily visible to you. I have addressed your comments and revised your uploaded file carefully. I hope that I have modified and explained the manuscript clearly enough, and it is to your satisfaction. We thank your comments which helped us improve the quality of this paper.

Reviewer 2 Report

It is an excellent article with a focus on clean irrigation agricultural products. I suggest to explain better the models associated to be used under other scenarios  

Author Response

Response to Reviewer 2 Comments

Point 1: It is an excellent article with a focus on clean irrigation agricultural products. I suggest to explain better the models associated to be used under other scenarios. 

Response 1: Thank you for this helpful suggestion. This manuscript has been revised carefully. I have corrected the problems that the reviewers brought forward point by point and have replied carefully, addressing the problems raised. And all revisions are clearly highlighted by using the "Track Changes" function in Microsoft Word 2016, so that changes are easily visible to you. I have addressed your comments and revised your uploaded file carefully. I hope that I have modified and explained the manuscript clearly enough, and it is to your satisfaction. We thank your comments which helped us improve the quality of this paper.

It is that “The SBM (Slack-Based Measure) model based on unexpected output is a model for measuring efficiency that was first proposed by Tone [28] and that, can effectively solve the "crowded" or "slack" of input factors caused by the traditional data envelopment model (DEA) model based on radial and angle. However, the SBM model has the same problem as the traditional DEA model; that is, it is difficult to further distinguish the differences between efficient decision-making units for decision-making units with the efficiency of 1. Based on the SBM model, Tone [29] defined the Super-SBM model, which is a combination of the Super-DEA model and SBM model, that combines the advantages of both models. Compared with the SBM model, the Super-SBM model can further compare and distinguish the efficient decision-making units at the forefront. ” (Line 127-Line 139)

 “The classical econometric method assumes that regions are independent. However, the flow of production factors flow can promote the improvement of AWGE in regions through the demonstration effect. Therefore, it is necessary to use the spatial panel econometric model to test this effect. Based on this, the present study uses a spatial econometric model for empirical measurements.

The basic traditional spatial econometric models mainly include the spatial lag model (SLM), spatial error model (SEM), and spatial Durbin model (SDM). The SLM examines the spatial spillover effect caused by the spatial dependence of the variables, and the SEM examines the spillover effect of the impact of the error term in the adjacent areas on the regions [4]. Compared with the SLM and SEM models, the SDM model considers the spatial correlation of dependent variables as well as the spatial correlation of independent variables and has both spatial auto-correlation and spatial interaction effects. At the same time, for endogenous problems, the SDM model can be used to obtain estimated values that are not biased by amplification.” (Line 179-Line 192)

Reviewer 3 Report

Authors have done an outstanding job by putting this manuscript together. The research is relevant and timely.

Author Response

Response to Reviewer 3 Comments

Point 1: Authors have done an outstanding job by putting this manuscript together. The research is relevant and timely. 

Response 1: We thank the reviewer for the overall very positive assessment of our manuscript. And all revisions are clearly highlighted by using the "Track Changes" function in Microsoft Word 2016, so that changes are easily visible to you. I hope that I have modified and explained the manuscript clearly enough, and it is to your satisfaction.  We thank your comments which helped us improve the quality of this paper.
